# Biotemplate Replication of Novel *Mangifera indica* Leaf (MIL) for Atmospheric Water Harvesting: Intrinsic Surface Wettability and Collection Efficiency

**DOI:** 10.3390/biomimetics7040147

**Published:** 2022-09-29

**Authors:** Edward Hingha Foday Jr, Taiwo Sesay, Emmanuel Bartholomew Koroma, Anthony Amara Golia Seseh Kanneh, Ekeoma Bridget Chineche, Alpha Yayah Jalloh, John Mambu Koroma

**Affiliations:** 1Key Laboratory of Subsurface Hydrology and Ecological Effects in Arid Region of the Ministry of Education, Chang’an University, Xi’an 710064, China; 2Department of Environmental Engineering, School of Water and Environment, Chang’an University, Xi’an 710064, China; 3Faculty of Education, Eastern Technical University of Sierra Leone, Combema Road, Kenema City 00232, Sierra Leone; 4School of Highway, Chang’an University, Xi’an 710064, China; 5Department of Geography-Environment and Natural Resources Management, Faculty of Social and Management Sciences, Ernest Bai Koroma University of Science and Technology, Magburaka City 00232, Sierra Leone; 6School of Economics and Management, Chang’an University, Xi’an 710064, China; 7Department of Environmental Engineering, School of Energy and Power Engineering, Xi’an Jiaotong University, Xi’an 710049, China

**Keywords:** atmospheric water, *Mangifera indica* leaf, biotemplate, polydimethylsiloxane, wettability, contact angle

## Abstract

Water shortage has become a global crisis that has posed and still poses a serious threat to the human race, especially in developing countries. Harvesting moisture from the atmosphere is a viable approach to easing the world water crisis due to its ubiquitous nature. Inspired by nature, biotemplate surfaces have been given considerable attention in recent years though these surfaces still suffer from intrinsic trade-offs making replication more challenging. In the design of artificial surfaces, maximizing their full potential and benefits as that of the natural surface is difficult. Here, we conveniently made use of *Mangifera indica* leaf (MIL) and its replicated surfaces (RMIL) to collect atmosphere water. This research provides a novel insight into the facile replication mechanism of a wettable surface made of Polydimethylsiloxane (PDMS), which has proven useful in collecting atmospheric water. This comparative study shows that biotemplate surfaces (RMIL) with hydrophobic characteristics outperform natural hydrophilic surfaces (DMIL and FMIL) in droplet termination and water collection abilities. Water collection efficiency from the Replicated *Mangifera indica* leaf (RMIL) surface was shown to be superior to that of the Dry *Mangifera indica* leaf (DMIL) and Fresh *Mangifera indica* leaf (FMIL) surfaces. Furthermore, the wettability of the DMIL, FMIL, and RMIL was thoroughly investigated, with the apices playing an important role in droplet roll-off.

## 1. Introduction

Water shortage has become a global threat to the growth of humans particularly in the desert and xeric areas [1,2]. Floras and faunas have greatly suffered from this adverse environmental impingement as its effect is lucid around the world with tropical countries suffering the most. The situation has been exacerbated by climate change impact, greenhouse gases, and lack of access to drinking water thereby intensifying poverty, diseases, and natural disaster [3]. Atmospheric water collection offers a propitious remedy to clean water in the affected regions and the world at large considering the ubiquitous nature of moisture in the atmosphere [4,5]. The atmosphere provides 50,000 km^3^ of water and accounts for about 10% of the world’s freshwater sources. This earth’s atmosphere holds water in the form of droplets, humidity, or vapor [6].

Despite the ubiquity of atmospheric water, vapor from the atmosphere is ephemeral and multi-scalar which requires a delicate surface control mechanism for effectively capturing, coalescence, and transportation. It is imperative to develop a systematic understanding of the individual contributions of these three main stages (capture, coalescence, and transport) for effective water collection. During the “capture” stage tiny droplets (1–40 μm) in diameter need to be captured from the air and these droplets are pushed by the wind and captured onto the solid surface by direct contact [7,8]. Even though the tiny water droplets are trapped by the surface, if they are not immediately collected owing to evaporation or wind, they may be lost back into the air. At the “Coalescence” stage, these tiny droplets merge during contact on the solid surface to form a single daughter droplet. During the “transport” stage the droplets will roll off the surface by gravity after they grow large enough to be harvested. Crucial to the collection include the surface composition of the collecting site, its area, and as well as the tilted angle of the surface.

According to the literature [9], droplet collection shows that patterned wettable surfaces exhibit both heat transfer coefficient and collection rate relative to unpatterned surfaces. Chen et al. [9] further took this concept by creating micro-pyramids with fluorinated nanograss texture to easily shed droplets off the surface. The created hydrophilic pillars caused the ejection of water droplets from the surface when adjacent droplets coalesced. Similar behavior was observed by Rahman et al. [10] on hydrophobic pillars coated with tobacco mosaic virus templated nanograss. Lastly, the Aizenberg group combined the macroscopic bump topography of the *Stenocara* beetle, a tapered diameter of cactus needles, and lubricant-filled pores of the pitcher plant. These complex patterns enhanced the flux of water to the surface, accelerated the motion of droplets to the collector, and provided a nearly frictionless surface for droplet sliding [11]. Natural species such as desert beetle [7,12], cactus [13,14], spider [15,16], green tree frogs (*Litoria caerulea*) [17], and *Nepenthes alata* [18,19], have evolved intelligent structural characteristics and distinct wettability for very efficient water harvesting from naturally occurring sources, which has inspired recent efforts to fabricate biobased surfaces for atmospheric water collection. For example, lotus leaf-like microstructure was fabricated using filter paper as a template to prepare super-hydrophobic polytetrafluoroethylene (PTFE) surfaces [20]. This is considered a very sustainable approach, considering the gradual depletion of traditional natural water sources due to climate change or other environmental-related factors. However, most functional surfaces are still hindered by the inherent trade-offs imposed by individual droplets. In that regard, it remains difficult to design artificial surfaces that have the maximum potential and benefits as natural surfaces. These benefits include; easy nucleation, frequent surface refreshing, and well-defined droplet shedding size under a wide range of environmental conditions, which are difficult to achieve with a designed artificial surface. In that direction, we implemented a facile replication method known as soft lithography, even though other methods like printing techniques and electrospinning have been used to mimic natural structures.

Some comparative studies on artificial surfaces (PDMS) and living leaves have been carried out with special reference to surface wettability. Soffe, R., et al. [21], compared artificial surface (PDMS) to living leaf to enable the identification of individual factors influencing microorganism function and viability in a controlled environment. PDMS replica leaf offers a control surface replica considering its inherent properties which were used to investigate microbe-microbe and microbe-plant interactions in the phyllosphere, which will enable mitigation strategies against pathogens to be developed. Organic surfaces, such as plant leaves, are more complicated than manufactured mould materials. Wang et al. [22] investigated the impact of leaf surface features such as epidermal wax, trichomes, and stomata on contact angle. Sun et al. [23] compared the wettability of three hydrophobic plant leaves with biomimetic production of an aluminum alloy’s superhydrophobic surface. Koch et al. [24] described a strategy for creating a low-cost, high-resolution reproduction technology employing organic and synthetic surfaces coated with wax crystals. The morphology and wettability of the underside of English weed (*Oxalis pescaprae*) leave and their biomimetic duplicates were studied by Pereira et al. [25].

Inspired by nature with special reference to the *Mangifera indica* tree (MIT), a plant commonly found in Africa and other tropical regions in the world with the potential to adapt and survive harsh climatic conditions has been used as a baseline for this study Figure 1a. India produces 57.18 percent of the total worldwide Mangifera Indica output of 19.22 million tonnes, making it the world’s biggest producer [26]. *Mangifera Indica* tree (MIT) can withstand low and high temperatures within the range of −39 °C to 42 °C) [27]. By narrowing this work to the functional elegancy of the surface of the *Mangifera Indica* leaf, we compared the water collection mechanism of three novel surfaces in a bid to determine the most efficient water harvesting surface. Even though MIL cannot be grown or found throughout the world but it can be used as a template to replicate water harvesting structures. In one of our recent works [28], we studied the wettability and droplet collection dynamics of MIL; it was understood that the combination of microgrooves, curvature, apex, and veins aided the coalescence and transportation of water droplets. The secret to such droplet movement lies in the unique structural feature of the surface as shown in Figure 1b,c.

In this study, MIL was artificially replicated with a polymeric material for the first time and its water collection efficiency was compared with natural leaves. With a well-defined aim of determining water collection efficiency among FMIL, DMIL, and RMIL, a detailed investigation of water droplet dynamics and the kinetics of coalescing water collection will be conducted; these facts will provide valuable insights, resulting in thorough apprehension of the water collection process among these samples. Similarly, comparative surface wettability will be monitored to provide a shred of convincing evidence to prove which of the surfaces (FMIL, DMIL, and RMIL) provide efficient atmospheric water collection. The results of this work will highlight the often-overlooked material in regards to it role it plays in atmospheric water harvesting. The effect of the droplet exit point called the “apex” will be carefully analyzed and thus understanding its overall contribution to water collection.

## 2. Experimental Section

### 2.1. Materials

Macklin Biochemical Co., Ltd., Shanghai, China, supplied the polydimethylsiloxane (PDMS) elastomer (Sylgard 184, Dow Corning). ANPEL laboratory Technologies in Shenzhen, China, provided the D20 humidifier while South Huadi Avenue in Guangzhou, China, provided the external environment test instruments (Anemometer and Hydrometer). The fresh *Mangifera indica* leaves were obtained from Lijiang, PR. China, as the adaxial surface was used for this study. Tianjin Rionlon Pharmaceutical Science & Technology Development Co., Ltd. (Tianjin, China) supplied the ethanol. The Scanning Electron Microscope (SEM) (S-4800 HITACHI, Tokyo, Japan), and Fourier transmission infrared (FTIR) spectroscopy (Perkin Elmer Spectrum Two, Waltham, MA, USA), were utilized for surface morphology and sample characterization. The contact angle (CA) and droplet motions were studied by OCA (JC 200D-1) and (VHX –900F) optical instruments, while Nikon D90 digital camera was used to video capture water droplets. An electronic scale (JY501), and ionized water were also utilized. A clear acrylic chamber was employed as a micro-weather station while Image-J and Origin 2018 were used for detailed investigation.

#### Leaf Samples

Three sample leaves were used with uniform sizes of 2 × 1 cm.

(i)Fresh *Mangifera indica* leaf (FMIL) was used with no further modification.(ii)The Dry *Mangifera indica* leaf (DMIL) was kept to be dried in the laboratory at a room temperature of 21 ± 3 °C for 150 days.(iii)Replicated *Mangifera indica* leaf (RMIL) was fabricated as described in Section 2.2.

The apices were thoroughly explored since they play an important role in the droplet’s roll-off, which is consistent with the findings presented by Ting Wang and colleagues [29].

### 2.2. Replication of Artificial MIL as Water Harvesting Substrate

To replicate the MIL, we used a facile soft lithography method similar to the one reported by Sharma, V et al. [30]. Polydimethylsiloxane (PDMS) elastomer (Sylgard 184, Dow Corning) was made by combining the base with a curing agent at a weight ratio of 10:1. FMIL was cut into medium size of about 4.5 × 2.0 cm. A mass ratio of 20:2 of PDMS (base and curing agent) was stirred for 15 min with a spatula and degassed with a desiccator to remove bubbles. The degassed substrate was poured into a flat tray and pre-cured in an oven at 70 °C for 5 min to get a molten substrate. The fresh MIL was pressed into the molten substrate to replicate the adaxial surface of the leaf. Before the final cure, the molten PDMS substrate with the submerged leaf was covered with glass and a 2 kg weight solid was placed on top to press the template leaf. The obtained substrate was fully cured at 90 °C for 1 h before finally peeling off the leaf to have a biotemplate surface as illustrated in Figure 2. The primary and, secondary veins were visibly replicated similar to the natural leaf. The replicated leaf measures 2 cm in width, 4.5 cm in length, and 3 mm in thickness. Before the water collection experiment, the surface of the samples was cleaned with ethanol.

### 2.3. Characterization

Scanning Electron Microscope SEM was used to examine the materials’ microstructure and morphological surface (S-4800 HITACHI) in Xi’an city, China. The water contact angle was measured with a (JC 200D-1) goniometer using a sessile droplet of 5 μL, and the silhouette image was recorded with an integrated camera. The chemistry of the samples was investigated with Fourier Transmission Infrared (FTIR) spectroscopy (Perkin Elmer Two Spectrum), while atmospheric moisture dynamics on the samples were studied using an optical microscope (VHX-900F).

### 2.4. Water Collection Setup

To quantify the water harvesting performance of the samples (FMIL, DMIL, RMIL) we instituted an experimental setup. The setup mimicked atmospheric water droplets at low velocity as it was done by installing commercial humidifier D20 which generated moist airflow perpendicular to the samples. The ability of the materials to collect droplets was tested using a customized experimental setup that included a clear acrylic container that served as a micro-weather station. Humidifier D20, thermometer, hygrometer, anemometer, and electronic scale are included in the micro-weather station. Within the chamber, the humidity and temperature were 85% and 23 °C respectively as the TH603A hygrometer was used. An anemometer was utilized to measure the atmospheric water velocity (2.2 ms^−1^), while the perpendicular distance in-between the samples and the humidifier was 7 cm. The collected water was measured using an electronic scale at 20-min intervals for two hours, and all samples were almost consistent in size (4.5 × 2.0 cm). A 100mL beaker was utilized to harvest water from the leaves, with a 10cm distance between the samples and the collecting beaker. The surface samples were carefully attached to the sample holder with an inclined angle of 45° (see Figure 3).

## 3. Result and Discussion

### 3.1. Surface Wettability of Novel MIL

To understand the correlation between the surface roughness of novel MIL and water droplets, it is imperative to understand the hiding treasure and theories in the study of wettability concerning the topic under review. Surface wettability is strongly related to surface roughness and intrinsic material properties. This has led to so many theoretical paradigms advanced by scientists such as; Thomas Young [31,32] who propounded the concept of contact angle (CA) of a liquid and in 1805 developed the Young equation. Greenspan [33] found moving droplets on a wettability gradient surface, whereas Brochard [34] focused on the isothermal situation and thought that gravity had no effect on droplet structure. Subramanian et al. [35] achieved the driving force acting on a spherical-cap droplet traveling over a wettability gradient surface and predicted the droplet moving velocity. When the intrinsic water contact angle (θ) on a flat solid surface is larger than the Intrinsic wetting threshold (IWT), a hydrophobic surface is obtained. Conversely, when the intrinsic water contact angle (θ) on a flat solid surface is smaller than the Intrinsic wetting threshold (IWT), a hydrophilic surface is obtained. The intrinsic wetting threshold (IWT) of water determines surface roughness and surface chemical compositions thus serving as a boundary between hydrophilic and hydrophobic when the liquid is deposited. The relevant characterization indicators are contact angle (CA), static angle (SA), or contact angle hysteresis (CAH). The CAH or SA indicates the difference between the advancing and receding angle based on the activated energy required for the movement of a droplet while CA defines the degree of water repellency of liquid on the solid surface. The dynamic wettability is defined by the equation below [36].
(1)$$\mathrm{mg}\;\sin\alpha =\sigma w\left({\cos\theta }_{r}-{\cos\theta }_{a}\right)$$
where m and σ represent weight and surface tension respectively while w, g, $\theta_{a}{\;\mathrm{and}\;\theta }_{r}$ represent the width of the liquid droplet in contact with the surface, gravitational acceleration, advancing contact angle, and receding contact angle respectively. The relationship between solid rough surfaces and contact angles $\left(\theta_{a}\right)$ can be defined by the Koch curve fractal formula [37,38]
(2)$$\cos\theta_{a}=f_{1}{\left(\frac{L}{l}\right)}^{D-2}$$
where $\cos\theta =f_{1}.f_{2}$ represent a fraction of air surface under the droplets, as $f_{1}+f_{2}=1$. L and l are the upper- and lower-limit scales of the surface, respectively, and D is the fractal dimension. The value of L/l has close ties with apparent contact angle and surface roughness thus the roughness exhibit nanoscale or microgroove structure that promotes wettability. Based on the above literature, we determine the wettable properties of the control experimental samples (the Fresh, Dry, and Replicated MILs). The apparent contact angles are shown in Figure 4 with details in Table 1. It is important to note that the contact angle of both fresh and dry MILs exhibited low values representing hydrophilic features (Figure 4a,b) while the replicated MIL exhibited a high value accounting for its hydrophobicity (Figure 4c). From observation, it can be concluded that the surface microgrooves of FMIL were prone to absorption as a succulent plant compared to the DMIL and RMIL. This procedure is critical for determining the wetting characteristics of MILs.

Figure 5a,b show microscopic photos (VHX-900F) as well as a schematic depiction of droplet activity on the materials. To further understand the droplet channel behavior, we schematically demonstrated the behavior of the droplets on the functional surfaces (FMIL, DMIL, and RMIL). Taking measurements of the coalescence and channelling of droplets on the samples was rather challenging as the droplets spreading and disappearance were fast. The collected droplets formed a water film on the FMIL in 3 s during droplet coalescence while it took 7 and 10 s to form water film on DMIL and RMIL respectively Figure 5a. The tangential sweeping behavior of the coalesced droplets on the samples was ignited by a tilting angle of 45°. One intriguing feature observed was the efficient droplet channel behavior exhibited by RMIL surface compared to FMIL and DMIL. The RMIL surface exhibits excellent transport behavior with a time limit of 1.10 s from the droplet capture site to the apex point of dripping while the DMIL and FMIL exhibited a time limit of 1.29 s and 1.43 s. This discovery is consistent with the findings of X. Liu and P. Cheng [39], who state that a hydrophilic surface prefers water to a hydrophobic surface.

We further investigated the effect of the apex of the functional surfaces during the water collection process. The apices were shaped triangularly with a distinct threshold water volume of 50 μL droplets. The FMIL apex showed the lowest droplet volume on the surface as a huge volume of the droplet was lost to absorption and ambient air as compared to dry and replicated MILs which were only affected by ambient air. The curvature shape of FMIL and DMIL morphologies of this novel species is different from the RMIL, thus playing a key role in the drainage behavior at an inclination angle of 45°. Similarly, the droplet shapes on the DMIL and FMIL were oval and dumbbell in shape respectively while the RMIL shows an elliptical shape. Three major forces influenced the coalescence and transportation of a droplet on a solid surface, these forces include; coalescence driving force (F_D_), hysteresis force (F_H_), and wettable gradient force (F_W_). These forces help dictate and describe droplets’ interaction with the solid surface in motion [40,41]. At this stage as demonstrated in Figure 5b, the tilted angle and gravitational force overpowered the retention force, aiding atmospheric moisture collection.

Droplet dynamics and interaction with the functional surfaces show that the initial droplets on the FMIL rapidly grew to a critical size compared to the DMIL and RMIL. The droplets on the DMIL and RMIL were stretched along the microgrooves and the gutter-like surface on the respective samples. The intriguing phenomenon observed on the RMIL shows the liquid column being attracted by the gutter-like surface as the growth of the droplets were confined and dictated by the size and shape of the gutter-like feature on the surface, which led to the quick transportation of water film to the apex.

We further examined the surface roughness of the samples in terms of the surface’s microgroove depth, width, and length of the samples as shown in Table 2. The visualized images of FMIL, DMIL, and RMIL show a microgroove surface depth of approximately 20–22 μm, 13–15 μm, and 18–20 μm respectively. Probably, the in-depth cavity of microgrooves on the FMIL surface is one of the few reasons for the quick dissipation of droplets compared to DMIL and RMIL which agrees with the view propounded by Comanns and group [42]. In this study, the mean height of the surface sharpness with respect to the reference plane is known as surface roughness (Sr). To practically define and compare the surface roughness (Sr), we 3D profiled (Figure 6) and calculated the surface morphology of the three (3) samples (FMIL, DMIL, RMIL).

The following surface roughness average (Sr) values were obtained; FMIL = 20.4 μm, DMIL = 14.7 μm and 11.06 μm. These values suggest that FMIL has the roughest surface followed by DMIL surface while RMIL possesses a relatively less rough surface compared to the afore surfaces. It can be concluded that FMIL demonstrates more affinity for water droplets (Figure 6a), while DMIL demonstrates a little affinity for water droplets (Figure 6b). The latter (RMIL) demonstrates very little or no affinity for water droplets (Figure 6c). During microscopic examination of the three (3) samples, the projected veins on the adaxial surfaces especially the dry and replicated samples play a key role in the movement of droplets for efficient water harvesting.

### 3.2. Atmospheric Water Transportation and Collection Efficiency

As seen in Figure 3, the MIL samples (DMIL, FMIL, and RMIL) were attached to the sample holder with an inclination angle of 45° and directly perpendicular to the commercial humidifier. All samples are of uniform sizes while the size of the sprayed droplets by the commercial humidifier ranged from a few micrometers (μm) to millimeters (mm). At a room temperature of 21.0 ± 3.0 °C and with a relative humidity of 80–85%, the experimental setup for atmospheric water collection was instituted. Atmospheric water harvesting is a water collecting method in which mist droplets are absorbed and deposited on a surface without heat transfer [43]. This startling invention was noticed experimentally using the digital D90 Nikon camera. To ascertain the water collection efficiency of the samples, we simply quantify the water collection efficiency of each surface although capillary adhesion played a “holding back” role in the drainage process, especially with the replicated leaf (RMIL). For this study, we define the capillary adhesion force as:(3)$$P_{\mathrm{cap}}=\pi \mathrm{Dwy}\;\left(1+\cos\theta r\right)$$
where Dw denotes contact diameter, θr is the receding contact angle, and y represents the surface tension of water. The $P_{\mathrm{cap}}$ defines the capillary adhesion force while pi(π) = $\frac{22}{7}$ [44]. When the force of gravity (F) surpasses the capillary adhesion force, the critical droplet size on the surface that determines droplet detachment is achieved. This may be expressed mathematically as ρgV~πDwy (1+cosθr); where (V) is droplet volume, (g) is the acceleration due to gravity, and (ρ) is the water density. The dynamics of droplet coalescence behavior on both DMIL and RMIL are much slower compared to FMIL, similarly, the dissipation process of the droplet to either absorption or evaporation is much faster with the FMIL than both DMIL and RMIL. Inversely, the transportation duration of the water film from the point of coalescence to the apex for drip-off is much faster with RMIL when compared with DMIL and FMIL (RMIL > DMIL > FMIL). The RMIL surface exhibits a hydrophobic value of CA = 104° while DMIL and FMIL exhibit hydrophilic features with CA of 66° and 80° respectively. The effect of adhesion force between droplets on the surface was also reviewed as shown in Figure 7a. This effect creates adhesion on the surface as the droplets tend to move towards the apex and this process is known as the droplet pinning effect as shown in Figure 7a(i). The CA value of RMIL displayed a large repulsive force towards water describing this phenomenon as ‘RMIL hates water” (Figure 7a(ii)). The surface dynamic behavior reaction displayed is similar to the Lotus leaf effect with self-cleaning and water bouncing ability [45]. The FMIL and DMIL surface microstructures as seen in the SEM images greatly helped droplet interaction with the functional surfaces.

Among the three water collection samples used for this study, the RMIL surface-displayed efficient water collection with a total of 7.02 gcm^−2^h^−1^ compared to DMIL and FMIL surfaces with a total water collection of 6.20 gcm^−2^h^−1^ and 5.89 gcm^−2^h^−1^ respectively as shown in Figure 5b above. According to reported works [43,46], hydrophilic surfaces have been reported to display better water collection efficiency than hydrophobic surfaces. On the contrary here, the hydrophobic RMIL surface displayed better water collection efficiency than the hydrophilic surfaces (DMIL and FMIL). The hydrophilic DMIL and FMIL surfaces have a lower water collection efficiency as this is possibly due to the surface’s capacity to hold absorbed water molecules within its microgrooves when compared to the hydrophobic RMIL surface.

Figure 7b demonstrates the macro view of droplets on the functional surfaces with naked eyes. Captured droplets on the FMIL coalesce with neighboring droplets faster than DMIL and RMIL; as this phenomenon is similar to the VHX microscopic study in the above literature. Water droplets quickly spread across the FMIL surface within 28 s (t = 28 s), and the dispersed water droplets got connected to create a water film while some amount was lost by absorption, ambient air, and other environmental parameters (Figure 7b(i)). Similarly, on the DMIL surface, individual droplets take 36 s (t = 36 s) to merge and form relatively large droplets as shown in Figure 7b(ii). This relatively large droplet covers the shallow microgrooves on the surface, thus giving less opportunity to adhesion force to resist gravity as water roll off at an inclination angle of 45°. In contrast with the above surfaces (FMIL and DMIL), on the hydrophobic RMIL surface, individual tiny droplets are retained on the surface and directed towards a more wettable region as shown in Figure 7b(iii). For this surface, 58 s (t = 58 s) is needed to form a liquid bridge and with the help of tilt angle (45°) and gravity, efficient water is collected. During the transportation process, several liquid columns on the RMIL surface were transported as tails of the liquid bridge droplets, this helped refresh the surface for a new collection cycle. Droplets on the hydrophobic RMIL surface are forced to sip into the lower valley region described here as an artificial vein and microgrooves assigned with a yellow dot line and small bubbles in Figure 1d. At this point, the Laplace pressure gradient is generated thus enhancing the directional movement of water droplets while in the Cassie state [47].

According to Figure 8, the efficiency of harvested water was in consonant with the time the atmospheric water was sprayed to be collected from the samples. The more time the water was sprayed on the surfaces, the more water was collected. Despite the 20 min time interval in the water collection cycle, the most efficient water collection from the three samples was determined in 2 h. As seen in Figure 5b above, 5.89 gcm^−2^h^−1^ was the total amount of water collected for FMIL, while DMIL and RMIL accounted for 6.20 gcm^−2^h^−1^ and 7.02 gcm^−2^h^−1^ respectively. To ascertain the amount of harvested water, an electronic scale was utilized to measure the quantity harvested. One intriguing phenomenon observed from RMIL was its dual role in the collection process; (i) It displays great adhesion force which helps to capture droplets and; (ii) It undermines the fast release of the droplet to be transported to the apex for quick water collection as compared to FMIL and DMIL. The advantage FMIL and DMIL have over RMIL is the fast nucleation of the droplet with little or no space in between neighboring droplets as compared to RMIL. The merged water droplet travels in a direction with a more wettable gradient, causing coalescence driving force (F_D_) to increase, while gravity, curvature, and the effect of the Laplace gradient aid in the drip off. The ratio of captured water on the functional surface to the total transported water to be harvested is directly proportional to the quantity of water collected in situ. Based on the above analysis, the result in Figure 8 demonstrates that the FMIL with more microgrooves depth exhibited good water holding ability with very low water collection efficiency compared to the latter. The result is consistent with DMIL which also exhibited lesser water collection efficiency with lower microgroove depth, width, and length. However, the RMIL which has a relatively smoother surface with a gutter drainage-like feature displayed a superior transportation and water collection efficacy than the other samples. Furthermore, the size of the dripping droplets from the replicated surface (RMIL) was larger than both the FMIL and DMIL. However, a detailed water collection comparison of these surfaces (FMIL, DMIL, and RMIL) with other reported surfaces is given in Table 3.

### 3.3. Characterization

The surface composition of FMIL, DMIL, and RMIL samples was investigated using a scanning electron microscope (SEM) to better understand the surface morphology. Figure 9a depicted a rough mesh-like surface that aided in droplet gathering and pinning. The first step in the growth of an individual droplet is to fill the cavities of the rough mesh-like structure before they begin to extrude upwards. In Figure 9b, only the protruded veins are visible, with little or no cavities or ridge-valley-like features. At this stage, there is little filling of cavities; rather, the protruded veins assist in transporting droplets, especially at an inclination angle during collection. The replicated surface (RMIL) in Figure 9c has a gutter-like surface than the other samples (FMIL and DMIL). Droplets on this surface (RMIL) are easier to be transported than others, but they take a longer time to be merged to form a large water film, which aids in initiating droplet movement for collection. Similarly, due to its surface nature, it discourages droplet absorption during the collection process, and as such a reasonable volume of water was channeled for collection as shown in Figure 5 and Figure 8 respectively.

The infrared (IR) spectra of the individual samples showed distinct IR bands, as illustrated in Figure 9d. The FTIR spectra of the three samples were determined with peaks belonging to the hydroxyl group (O-H) spanning from 3911 to 3132 cm^−1^. The FTIR spectrum for the FMIL (3411 cm^−1^) is similar to that of RMIL (3432 cm^−1^) while DMIL was 3911cm^−1^ with a phenols functional group. The peaks at 2921 cm^−1^, 2910 cm^−1^, and 2367 cm^−1^ resulted from both asymmetrical and symmetrical stretching vibrations and are assigned to the (C-H) functional group. In comparison, there is an increase in the (O-H) group for DMIL than for both FMIL and RMIL, whereas the (C-H) group increase for FMIL and DMIL (2921 cm^−1^ and 2910 cm^−1^ respectively) than for RMIL (2367 cm^−1^). Furthermore, bands that are consistent with C=C stretching vibration range between 1675–1600 cm^−1^. The (C-H) bending alkane group produced peaks between 1465 and 1400 cm^−1^ which is consistent with RMIL. However, these findings revealed that the samples had little variation.

## 4. Conclusions

During water harvesting, the readily captured moisture occurs via a direct collision with the surface and thus an efficient water collection is harnessed which is influenced by a large receding contact angle. This is one of the main determining factors for the water harvesting performance from a functional surface. In this study, we experimentally investigated the atmospheric water collection efficiency from three (3) wettable surfaces (FMIL, DMIL, and RMIL). Among the three surfaces, the soft lithography method was used to successfully replicate one of the samples using fresh *Mangifera indica* leaf as a template. Our results show that the water collection performance of a functional surface hinges on many parameters and among which are; surface wettability, curvature, inclination angle, leaf veins, and the amount of moisture captured and transported to the apex for final collection. Our findings show that a defined wettable surface with a hydrophilic feature hinders efficient water collection as compared to a less wettable surface with a hydrophobic feature. Here, the hydrophobic RMIL surface displayed better water collection efficiency than the hydrophilic surfaces (DMIL and FMIL). The reduced efficiency of water collection from the hydrophilic DMIL and FMIL surfaces could be attributed to the surface’s ability to trap the absorbed water molecules inside its microgrooves when compared to the hydrophobic RMIL surface. Overall, our findings demonstrate that replicated *Mangifera Indica* leaf (RMIL) surface with hydrophobic properties is preferred for effective water collection over hydrophilic Dry and Fresh *Mangifera indica* leaf (DMIL and FMIL) surfaces. It is also expected that this discovery can help offer insight into the fabrication of a moisture harvesting surface by utilizing a facile method while also assisting in the alleviation of water crises, particularly in tropical countries.

## Figures and Tables

**Figure 1 biomimetics-07-00147-f001:**
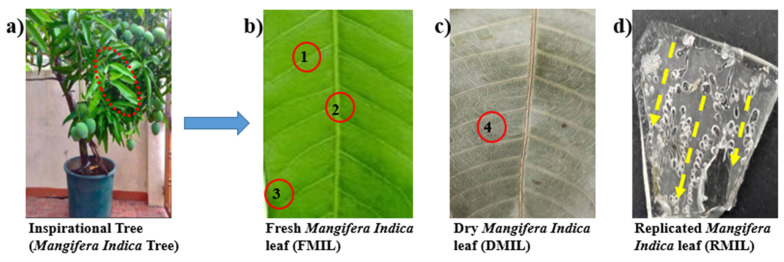
Photograph images (**a**) inspired *Mangifera Indica* tree (MIT) (**b**) Fresh *Mangifera Indica* leaf (FMIL); red cycle 1—Secondary vein, red cycle 2—Primary vein, red cycle 3—Curvature (**c**) Dry *Mangifera Indica* leaf (DMIL); red cycle 4—microscales, and (**d**) Replicated *Mangifera Indica* leaf (RMIL)-Yellow dot lines represent droplet drainage paths.

**Figure 2 biomimetics-07-00147-f002:**
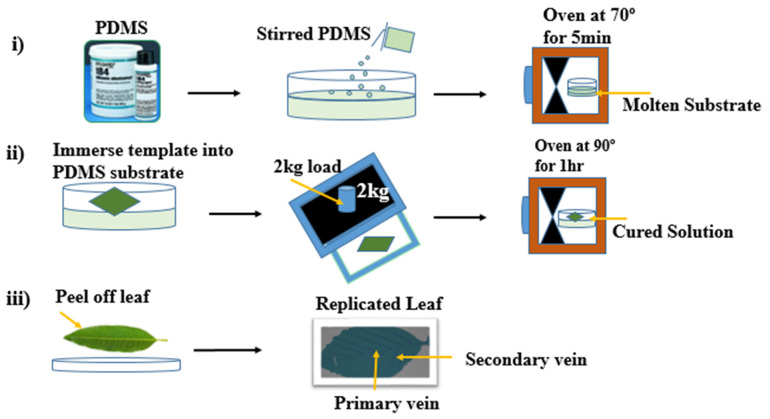
Soft lithography process for replication of plant leaf substrates using PDMS (**i**) Molten substrate preparation (**ii**) Template immersion into the molten substrate and (**iii**) Final replication stage.

**Figure 3 biomimetics-07-00147-f003:**
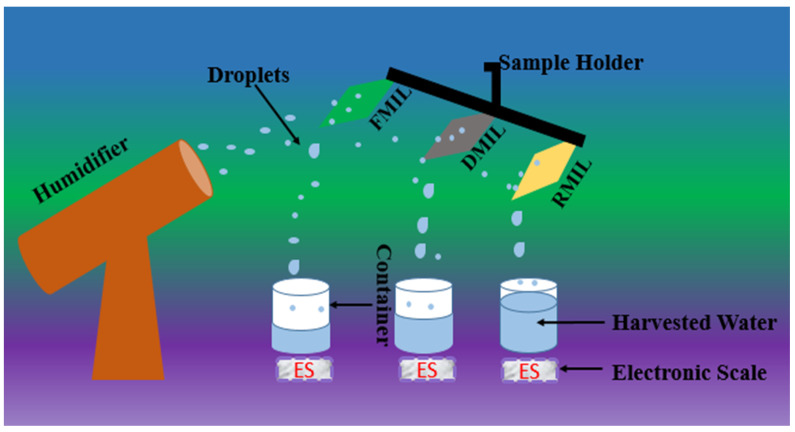
A schematic illustration of water collection experimental setup.

**Figure 4 biomimetics-07-00147-f004:**
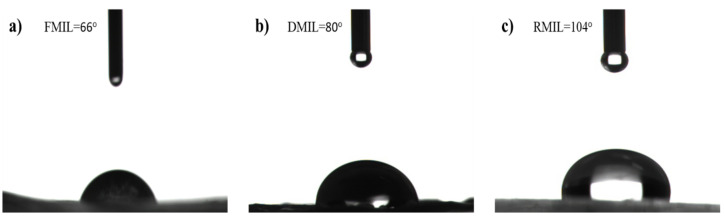
Contact angle (CA) of functional surfaces with a sessile droplet on; (**a**) Fresh *Mangifera Indica* leaf (FMIL) (**b**) Dry *Mangifera Indica* leaf (DMIL), and (**c**) Replicated *Mangifera Indica* leaf (RMIL).

**Figure 5 biomimetics-07-00147-f005:**
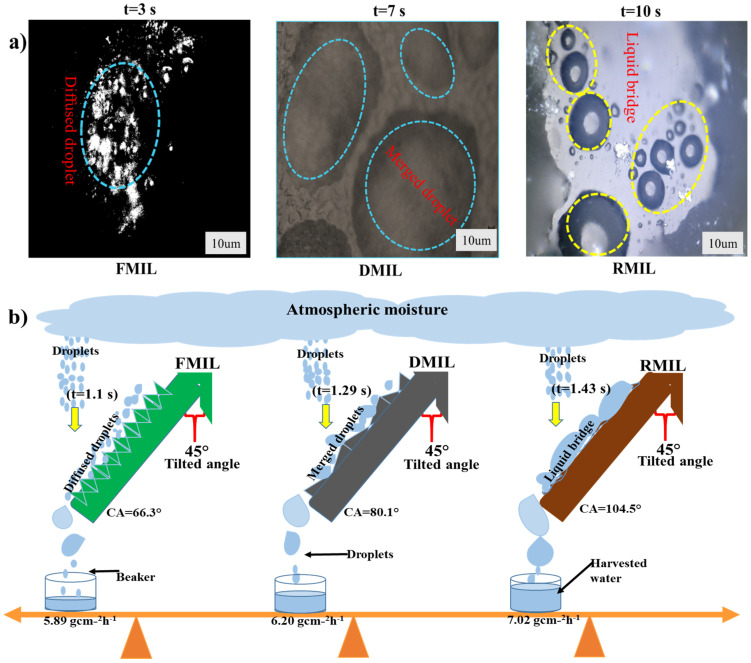
Microscopic and schematic illustration of the atmospheric water collection on various wettable surfaces (**a**) VHX- images displaying the coalescence behavior with respect to time and (**b**) the transportation duration for collection with respect to time; namely; Fresh *Mangifera indica* leaf (FMIL), Dry *Mangifera indica* leaf (DMIL), and Replicated *Mangifera indica* leaf (RMIL).

**Figure 6 biomimetics-07-00147-f006:**
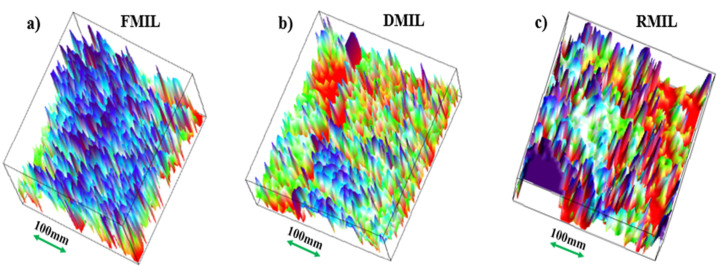
Surface morphology of the MIL: (**a**) Fresh *Mangifera indica* leaf (FMIL) (**b**) Dry *Mangifera Indica* leaf (DMIL), and (**c**) Replicated *Mangifera indica* leaf (RMIL).

**Figure 7 biomimetics-07-00147-f007:**
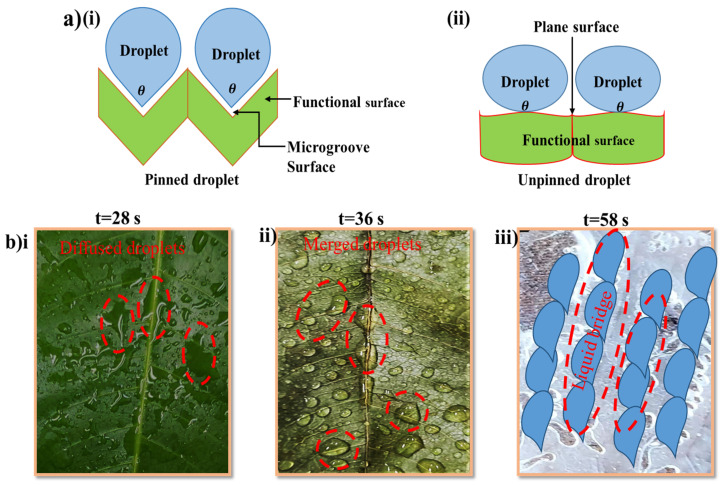
Schematic and photographic images of droplet behavior on functional surfaces; (**a**(**i**)) pinning and (**a**(**ii**)) unpinning effect of the droplet on a solid surface. (**b**) Coalescence and transportation behavior of droplet on solid surfaces with respect to time (**i**) Fresh *Mangifera indica* leaf (FMIL), (**ii**) Dry *Mangifera indica* leaf (DMIL), and (**iii**) Replicated *Mangifera indica* leaf (RMIL).

**Figure 8 biomimetics-07-00147-f008:**
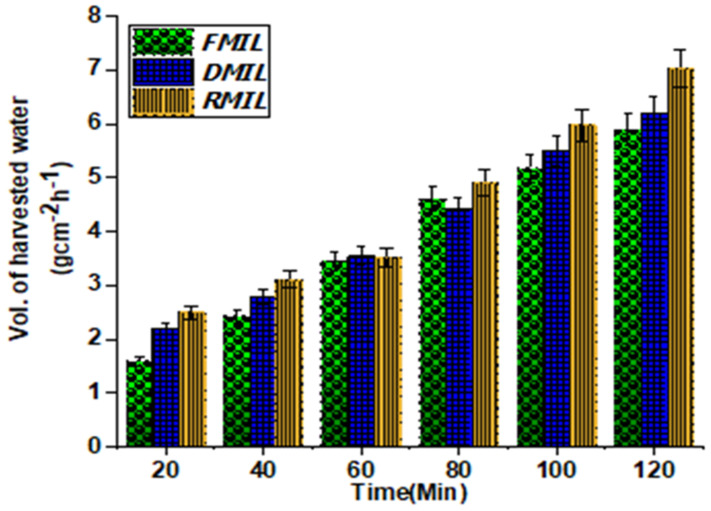
Water collection process from different functional surfaces. Amount of water collected by surfaces exposed to atmospheric water flow for 120 min with 20 min intervals for each cycle.

**Figure 9 biomimetics-07-00147-f009:**
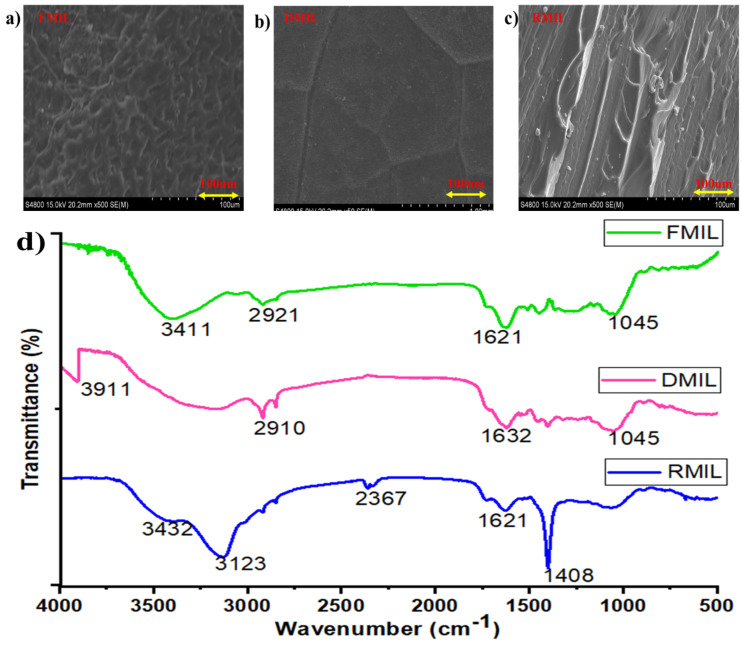
(**a**–**c**) SEM surface morphology of FMIL, DMIL, and RMIL, (**d**) FTIR spectra of 3 samples used during water collection; Fresh *Mangifera indica* leaf (FMIL), Dry *Mangifera indica* leaf (DMIL), and Replicated *Mangifera indica* leaf (RMIL).

**Table 1 biomimetics-07-00147-t001:** Surface wettability.

Materials	Surface Feature	Contact Angle (θ)	Advancing Contact Angle (θ_a_)	Receding Contact Angle (θ_r_)	Contact Angle Hysteresis (CAH)
FMIL	Hydrophilic	66 ± 2°	78 ± 2°	64 ± 2°	14 ± 2°
DMIL	Hydrophilic	80 ± 2°	91 ± 2°	77 ± 2°	13 ± 2°
RMIL	Hydrophobic	104 ± 2°	115 ± 2°	99 ± 2°	15 ± 2°

**Table 2 biomimetics-07-00147-t002:** Characteristics of functional surfaces.

Characteristics	FMIL	DMIL	RMIL
Microgroove depth	20–22 μm	13–15 μm	18–20 μm
Microgroove length	9–12 μm	10–15 μm	15–26 μm
Microgroove width	8–10 μm	9–12 μm	11–13 μm

**Table 3 biomimetics-07-00147-t003:** Water collection comparison of FMIL, DMIL, and RMIL with other reported surfaces.

Surfaces	Vol of Harvested Water (gcm^−2^h^−1^)	Material	References
*Nepenthes alata* surface	2.58	Natural leaf	[48]
*D. marginata* surfaces	0.72	Natural leaf	[49]
Biomimetic surface coatings	3.40	Polystyrene	[8]
Microstructured surfaces and mesh	0.18	Epoxy and Polyolefin	[50]
Fresh *Mangifera indica* surface (FMIL)	5.89	Fresh Natural leaf	This work
Dry *Mangifera indica* surface (DMIL)	6.2	Dry natural leaf	This work
Replicated *Mangifera indica* surface (RMIL)	7.02	Polydimethylsiloxane (PDMS)	This work

## Data Availability

Not applicable.

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
