# Peer review of "Biotemplate Replication of Novel Mangifera indica Leaf (MIL) for Atmospheric Water Harvesting: Intrinsic Surface Wettability and Collection Efficiency"

_biomimetics, 2022, doi:10.3390/biomimetics7040147_

Round 1

Reviewer 1 Report

The authors reported a method to collect atmosphere water using biotemplate replication of novel Mangifera Indica leaf. The experimental results seem to be interesting. However, the authors should clarify the following problems.

1. The Introduction should be improved to be more concise.

2. Page 7, Line 259: Please check the equation of cos(theta)=f_2.

3. Page 7, Line 273-275: It might be better to use droplets with volume of only several microliters for measuring the contact angle. 10 microliters seem a little larger for in such a case, gravitational effect might not be neglected.

4. Page 7, Line 276: It is better to give the data of contact angles with a deviation. How were the advancing and receding contact angles measured? The authors should describe it in the manuscript and cite some important papers. Besides, the data seem to has a much higher precision. As an example, it seems better to use 80 degrees rather than 80.1 degrees.

5. Page 8, Figure 5: The authors used three different surfaces for the investigation of water storage efficiency of RMIL. However, it seems incomplete because of the lackage of the data for water storage on a bare PDMS surface.

6. Page 10, Line 355-357: Please check Eq. (3) and the specification for each symbol.

7. Page 12, Line 418-420: In “As seen in Figure 5b above, ….collected for DMIL and RMIL respectively.”, are the data all an average or instantaneous coefficient for water storage? How were these data obtained?

8. Page 12, Figure 8: Please check this figure, which did not support the finding “ the efficiency of harvested water was in constant with the time the atmospheric water was sprayed to be collected from the samples” in Line 414-415, Page 11. Moreover, it is better to add error bars for the data in Figure 8.

9. It is better to compare water storage efficiency using the surfaces in this manuscript with the data in other literature.

10. Please improve the English of the manuscript.

Author Response

All answered concerns are bolded as seen in the attached document 

Reviewer 2 Report

The availability of fresh water is an increasingly important issue facing planet earth due to the effects of climate change. Harnessing water from the atmosphere is an attractive means to supplement the water shortages in arid and desert regions of the world. This study compares the atmospheric moisture harvesting efficiencies of a natural leaves FMIL and DMIL with that of its mimic the RMIL. They determined the factors that influences the performances and characterized their surfaces to shed light on the mechanism. Here are questions based on the presented work.

1.       Figure 1d, the image of replicated Mangifera Indica leaf (RMIL) has many bubbles. Explain why?

2.       Figure 4c, the image of sessile droplet for RMIL displayed appears to have CA of <90 degree, but it is reported to have CA of 104 degree. Explain why?

3.       Figure 9, the FTIR spectrum of RMIL shows many spikes, it is due to air bubbles.

4.       If RMIL is hydrophobic then why there are more –OH groups and less C-H groups than FMIL & DMIL. In fact the reverse is expected.

5.       Figure 6, what equipment is used to record the surface morphologies of FMIL, DMIL and RMIL and how the topologies and roughness evaluated.

6.       A brief comparison of the results in this work with that of the desert beetle and /or cactus will be helpful.

7.       Images for RMIL presented in Figure 1d and Figure 7biii are very similar, how?

8.       Which wetting mechanisms the systems FMIL, DMIL and RMIL follow based on the CA, SA, and CAH data.

9.       Is lotus leaves better then FMIL in harvesting water from the atmosphere?

Author Response

(The authors gave the same response as above.)

Reviewer 3 Report

The article wrote by E. Hingha Foday Jr. et al. entitled: « Biotemplate replication of Novel Mangifera Indica leaf (MIL) 2 for atmospheric water harvesting: Intrinsic surface wettability 3 and collection efficiency » present a work of great interest. The strategy described that allows to rapidly prepare original surfaces with harvesting properties is very interesting. The results seem to demonstrate a significant improvement of the artificial surfaces compared with the model. Unfortunately this work suffer of several drawback that makes this conclusion unclear. My opinion is that this article should be improved before being considered for publication in biomimetics.

Here find example of the drawbacks that need to be addressed:

1. The introduction is very long and contain a lot of generalisation without explaining the choice of the author for this model or alternative strategies described in the literature for water harvesting. The introduction needs to be rewritten.

2. L. 174, the sentence is not complete. I assume that the words:” was used with no further modification.” need to be added.

3. L. 234, Here is an important point, you describe that surface wettability is linked to the surface roughness and intrinsic material properties. This is perfectly true. As consequence, it is very difficult to compare dry leaf, fresh leaf and PDMS leaf wettabilities without considering the chemistry of there surface and consequently there surface energy. Fresh leaf should present more epicuticular wax than dry one and PDMS surface present a totally different composition. The authors should with this element before to conclude.

4. L. 243, The concept of super-hydrophobic surface is introduced. It should be explained.

5. Figure 4. Other images should be used, those one are to blurry and dark. Also the image s should be consistent in size and magnification to allow comparison. For the picture 5c, the apparent contact angle seem to be higher that 104° a better picture will dispel this doubt.

6. All numerical data (wettability and roughness) should be given with standard deviation to conclude on the significance of the difference between values.

7. Figure 5a, The coalescence of droplet is not clearly shown by the picture. Pictures of the same surface at different times should be more informative for the reader. Also the same time should be selected for all surfaces to allow the comparison for the reader. Scale bare should be more homogeneous.

8. Between L355 and 356, (3) is this an equation? I cant see the equal sign I don’t know what I am reading.

9. Figure 7b, I have the same comment than for the Figure 5a.

10. Figure 8. Here the standard deviation should be added before to conclude on this experiment.

11. SEM with higher magnification should to be performed. Micro-structurations may dramatically impact the wettability. Natural leafs (dry and fresh) should be compared with the leaf reproduction. SEM may be correlated with roughness measurment. This observation may be very interesting to understand the wettability.

12. Figure 9. I am not sure of the IR results. Of course observations on FMIL and DMIL give results very consistent with potential epicuticular wax. But IR for RMIL it is very different. Some of the bands look like artefacts (IR bands are not so angular, example 3559 cm-1) and should not be considered. From my point of view the observed spectra of RMIL should be done again to be usable. Here the IR of RMI looks like IR of residual epicuticular wax transferred during preparation and not washed by ethanol (not soluble) and not IR of PDMS. Additionally, I am not sure that IR provide any information consistent for this work.

Author Response

(The authors gave the same response as above.)

Round 2

Reviewer 1 Report

The authors have significantly improved the manuscript. The manuscript can be accepted for publication.

Author Response

The reviewer made no extra comment but rather accepted the manuscript for publication.

Reviewer 3 Report

The present manuscript has clearly been improved and as reviewer I must congratulate the authors for it. Even if improved, some point still need to be improved from my point of view.

One point was my previous third comment: to correctly understand the wettability the surface chemistry is important. It is unclear if we can compare PDMS surface and natural leaf. At list it should be more explained. A discussion on this point should be included.

The second point is the standard deviation. All numerical results should be presented with average error, not only some of them. For example, apparent contact angle should be presented with mean error.

From my point of view this points should be adressed before to be considered for publication.

Author Response

On page 2-3 with the green highlight, a comparative discussion was made relating to surface wettability of a replicated surface (PDMS) with a natural leaf surface. 

In table 1, a mean error value has been added to the apparent contact angle values.